# The Greater Proportion of Born-Light Progeny from Sows Mated in Summer Contributes to Increased Carcass Fatness Observed in Spring

**DOI:** 10.3390/ani10112080

**Published:** 2020-11-10

**Authors:** Fan Liu, Erin M. Ford, Rebecca S. Morrison, Chris J. Brewster, David J. Henman, Robert J. Smits, Weicheng Zhao, Jeremy J. Cottrell, Brian J. Leury, Frank R. Dunshea, Alan W. Bell

**Affiliations:** 1Rivalea Australia Pty Ltd., Corowa, NSW 2646, Australia; eford@rivalea.com.au (E.M.F.); rmorrison@rivalea.com.au (R.S.M.); cbrewster@rivalea.com.au (C.J.B.); dhenman@rivalea.com.au (D.J.H.); or rob.smits@australianpork.com.au (R.J.S.); 2Faculty of Veterinary and Agricultural Sciences, University of Melbourne, Parkville, VIC 3010, Australia; weichengz@student.unimelb.edu.au (W.Z.); jcottrell@unimelb.edu.au (J.J.C.); brianjl@unimelb.edu.au (B.J.L.); fdunshea@unimelb.edu.au (F.R.D.); 3Faculty of Biological Sciences, The University of Leeds, Leeds LS2 9JT, UK; 4Department of Animal Science, Cornell University, Ithaca, NY 14853, USA; alanwilliambell@gmail.com

**Keywords:** gestation, summer, sow, birth weight, fatness, in utero, fetal development, pig

## Abstract

**Simple Summary:**

Pig producers are required to supply consistent lean carcasses to the market. However, the pig production cycle contains seasonal variation in carcass fatness, such that pigs finished in spring have a greater carcass backfat thickness than those finished in summer. Our experiment showed that when sows were mated in summer they had an increased incidence of born-light progeny (≤1.1 kg), which when finished in spring, had increased fatness. This finding provides a novel explanation for the seasonal variation of carcass fatness and sets a research direction for future mitigation strategies.

**Abstract:**

The backfat of pig carcasses is greater in spring than summer in Australia. The unexplained seasonal variation in carcass backfat creates complications for pig producers in supplying consistent lean carcasses. As a novel explanation, we hypothesised that the increased carcass fatness in spring was due to a greater percentage of born-light progeny from sows that were mated in summer and experienced hot conditions during early gestation. The first part of our experiment compared the birth weight of piglets born to the sows mated in summer (February, the Southern Hemisphere) with those born to sows mated in autumn (May; the Southern Hemisphere), and the second part of the experiment compared the growth performance and carcass fatness of the progeny that were stratified as born-light (0.7–1.1 kg) and born-normal (1.3–1.7 kg) from the sows mated in these two seasons. The results showed that the sows mated in summer experienced hotter conditions during early gestation as evidenced by an increased respiration rate and rectal temperature, compared with those mated in autumn. The sows mated in summer had a greater proportion of piglets that were born ≤1.1 kg (24.2% vs. 15.8%, *p <* 0.001), lower average piglet birth weight (1.39 kg vs. 1.52 kg, *p <* 0.001), lower total litter weights (18.9 kg vs. 19.5 kg, *p =* 0.044) and lower average placental weight (0.26 vs. 0.31 kg, *p =* 0.011) than those mated in autumn, although litter sizes were similar. Feed intake and growth rate of progeny from 14 weeks of age to slaughter (101 kg live weight) were greater for the born-normal than born-light pigs within the progeny from sows mated in autumn, but there was no difference between the born-light and normal progeny from sows mated in summer, as evidenced by the interaction between piglet birth weight and sow mating season (Both *p <* 0.05). Only the born-light piglets from the sows mated in summer had a greater backfat thickness and loin fat% than the progeny from the sows mated in autumn, as evidenced by a trend of interaction between piglet birth weight and sow mating season (Both *p <* 0.10). In conclusion, the increased proportion of born-light piglets (0.7–1.1 kg range) from the sows mated in summer contributed to the increased carcass fatness observed in spring.

## 1. Introduction

Carcass fatness negatively affects the profitability of pig producers in some markets where a rigid specification on carcass fatness is imposed. In the Australian pig industry, backfat depth, an indicative measure of carcass fatness, is greater for pigs slaughtered in late winter and spring (July–October) than summer (January–March). The mechanisms for this phenomenon remain unclear. A study in Western Australia showed that less than 20% of the seasonal variation in backfat thickness was explained by the carcass weight [1]. The seasonality of carcass fatness variation creates complications for the pig industry to consistently produce lean carcasses to meet market needs. Understanding the factors contributing to the seasonality of carcass fatness variation would provide directions for developing mitigation strategies.

Pigs slaughtered during late winter and spring (July to October in the Southern Hemisphere) are the progeny born to the sows mated in summer (December to February in the Southern Hemisphere), and these sows usually experience hotter conditions during mating and early gestation. A recent climatically controlled study demonstrated that high environmental temperature occurring in the first half of gestation can increase fat deposition of the progeny of these sows [2]. Also, emerging evidence supports the role of gestational heat stress in fetal programming in pigs (reviewed by Johnson and Baumgard [3]), which provides a novel explanation for the greater backfat of progeny born to the sows mated in summer. However, a knowledge gap exists on whether the phenomenon of the fatter finisher pigs that are born to the sows mated in summer is a reflection of an increased proportion of born-light piglets, considering that born-light piglets have a propensity for increased adiposity [4,5,6]. Most climatically controlled studies had a small sample size (e.g., less than 20 L) and thus limited the likelihood for detecting impacts of gestational heat stress on piglet birth weights. Comparing piglet birth weights between sows mated in summer and autumn under natural conditions would allow us to understand the association between the frequency of born-light piglets and mating seasons of sows, which was conducted as the first part of the current experiment. In a natural seasonal pattern, the progeny pigs born to the sows mated in summer and autumn reach the slaughter weight (approximately 100 kg live weight in Australia) in spring and summer, respectively. The cooler environment during the spring finishing season usually allows greater feed intake than in summer, which may favor fat deposition, particularly in born-light piglets. To test this hypothesis, the second part of our experiment studied the growth performance and carcass composition in progeny stratified as born-light and born-normal groups from the sows mated in summer and autumn. We hypothesised that (1) the sows mated in summer have an increased proportion of born-light progeny than the sows mated in autumn, which may be associated with the hotter environmental temperatures experienced during early gestation; (2) born-light progeny from sows mated in summer are fatter when grown to a fixed slaughter weight, compared with the progeny from sows mated in autumn.

## 2. Materials and Methods

### 2.1. Animals and Experimental Design

The experiment consisted of two parts. Part 1 compared the birth weight of sows that were mated in summer with those mated in autumn. Part 2 followed a two by two factorial design to study the effects of the mating season of sows and the birth weight class (light vs. normal) of their progeny on growth performance and carcass traits of progeny at a fixed slaughter weight. A flowchart of the experiment is illustrated in Figure 1. The procedures that involved animals in the current study were in accordance with the Australian Code for the Care and Use of Animals for Scientific Purposes (8th edition, 2013), and the protocol (ID:17P074C) was approved by the Animal Ethics Committee of Rivalea Australia Pty Ltd., Corowa, NSW, Australia.

### 2.2. Housing Systems and Farrowing Outcomes of Sows

In Part 1 of the experiment, multiparous sows were mated in summer (February 2018 in the Southern Hemisphere) (*n =* 119; parity 3.7 ± 1.45 for mean ± standard deviation) and autumn (May 2018 in the Southern Hemisphere) (*n =* 118; parity 3.8 ± 1.50 for mean ± standard deviation) in a commercial Australian piggery (Rivalea Australia Pty Ltd., Corowa, NSW, Australia). The mated sows were housed in a pen of 40 until the sixth week of gestation, after which they were combined with another pen and housed in a pen of 80. All the sows were fed using electronic sow feeders while in the gestation pens, with each feeding station (40 pigs per feeding station) allowing individual feeding times. Sows were restrictively fed with an average of 2.2 kg diet per day until the 110th day of gestation to provide appropriate nutritional needs for their maintenance and pregnancy requirements. The sows mated in summer and autumn were housed in the same gestation pens using the same feeding system and allowance. The same gestation dietary formulation was used for the sows mated in the two seasons (Appendix A). Temperature loggers (DS1923 Hygrochron; OnSolution Pty, Baulkham Hills, NSW, Australia) recorded the air temperature inside the sheds every hour. The average daily temperature was calculated based on the hourly recorded air temperature. The average daily temperature during the first half of the gestation period was greater for sows mated in summer compared with those mated in autumn (Figure 2). On the 110th day of gestation, the experimental sows were moved to farrowing houses and then farrowed in individual farrowing crates. The number of piglets born-alive, still-born and mummified were recorded within the first 24 h post-farrowing before cross-fostering. Meanwhile, the new-born piglets were individually weighed using a digital scale. Birth weights of live piglets were recorded in all litters. Fifteen focal sows from each mating season were randomly chosen and weighed on the second-day post-mating (after summer lactation) and at the 110th day of gestation.

### 2.3. Physiological Monitoring and Injury Score of Gestating Sows

In Part 1 of the experiment, 15 gestating sows from each mating season were monitored for physiological signs of heat stress, including rectal temperature and respiration rate at 17:00 h one day per week until the 110th day of gestation. Rectal temperature was measured using a digital thermometer (Model DT-K01A; Liberty Health Products, Mount Waverley, VIC, Australia). Respiration rate was monitored by visually counting the number of chest contractions during a 30-s duration. Skin injury score was counted once per week from the second day of entry to the gestation pen until the 110th day of gestation using a modified version of the method described by Karlen, et al. [7]. The number of fresh scars resulting from injuries (scratches, abrasions and cuts) were counted, but older injuries (including abscesses) were not included in the week of counting.

### 2.4. Placental Weight Collection

Behavioural signs (i.e., nesting, pawing and chewing) were used to predict farrowing time in the sows. A rubber mat was placed behind the sows to catch placentae when a sow was about to farrow. A technician frequently checked the farrowing process and quietly put a soft coloured cable tie around the intact umbilical cord when a piglet was born and had been expelled from the sow. Once the cable tie was tied securely, the umbilical cord was cut between the cable tie and the piglet using a pair of sterile scissors, then the umbilical cord with the fixed cable tie was retracted into the sow. The piglet was weighed using a digital scale and the body weight was recorded to match the colour of the cable tie which was used for tagging the placenta. Placentae with the tagged umbilical cords were expelled in the middle of the farrowing process or could be as late as the completion of the farrowing process. Individual placentae were separated using the necrotised tissue edge as the reference border, then the tagged placentae were weighed using the digital scale and recorded to match the individual piglet. On average, five placentae and five piglets from each focal litter (*n =* 38 focal litters per mating season) were individually tagged, matched and weighed after birth.

### 2.5. Plasma IGF-1 in New-Born Progeny

Blood samples were taken via jugular venepuncture at 24 h post-farrowing from piglets that had suckled sows and received colostrum (*n =* 25 born-light and *n =* 27 born-normal piglets from sows mated in summer; *n =* 22 born-light and *n =* 20 born-normal from sows mated in autumn; balanced sex). Plasma samples were harvested after centrifugation at 2000× *g* at 4 °C for five minutes. Plasma IGF-1 concentrations were determined in singlicate and completed in one assay (Quantikine^®^ Human IGF-1 immunoassay kit, R and D Systems, Minneapolis, MN, USA).

### 2.6. Growth Performance and Carcass Traits of Progeny Pigs

In Part 2 of the experiment, 120 born-light (0.7–1.1 kg weight range; balanced sex) and 120 born-normal progeny piglets (1.3–1.7 kg weight range; balanced sex) from sows mated in each season were identified by individual birth weight. As a criterion for selecting the progeny pigs for Part 2 of the experiment, each litter contributed, at most, two born-light and two born-normal piglets. For evaluating growth performance of progeny pigs during the grower/finisher phase, pigs born to the sows mated in the two seasons were selected at 9 weeks of age based on similar body weight (20.4 ± 1.12 kg (mean ± standard deviation) for born-normal groups and 19.6 ± 1.12 kg (mean ± standard deviation) for born-light groups) as well as similar birth weight (1.5 ± 0.25 kg (mean ± standard deviation) for born-normal groups and 1.0 ± 0.25 kg (mean ± standard deviation) for born-light groups. Following the selection criteria, 56 born-light and 56 born-normal progeny pigs with balanced sexes were selected from each mating season and then housed in group pens during the grower phase (9–14 weeks of age) (*n =* 14 pens per birth weight class × mating season treatment; *n =* 4 pigs per pen; an equal number of pens per sex). During the finisher phase (from 14 weeks of age to 101 kg live weight), pigs were individually housed (*n =* 56 individual pigs per birth weight × mating season treatment). The sheds for grower and finisher phases were semi-climatically controlled and used fans, misting, water dripping system in summer and central gas heating in winter. Temperature loggers (DS1923 Hygrochron; OnSolution Pty, Baulkham Hills,, NSW, Australia) recorded the air temperature inside the sheds. The average overall temperature between 9 weeks to 14 weeks of age was 22.6 ± 2.89 °C and 25.0 ± 2.78 °C (mean ± standard deviation) for progeny born to the sows mated in summer vs. autumn. The average overall temperature between 14 weeks of age and slaughter was 23.3 ± 2.96 °C vs. 24.9 ± 2.73 °C (mean ± standard deviation) for the progeny born to the sows mated in summer vs. autumn.

Feed delivery and body weights were recorded weekly. Pigs were sent to a commercial abattoir when they reached 101 kg live weight. Hot carcass weights (Australian Trim 1 standard [8]: visceral tissues off, head on, trotters on), backfat thickness (P2 site: last rib, 6.5 cm from midline; Hennessey Chong’s probe method), and loin depth were recorded. The progeny born to the sows mated in summer finished between 24 October and 14 November 2018 (spring in the Southern Hemisphere), and the progeny born to the sows mated in autumn finished between 30 January and 8 March 2019 (summer in the Southern Hemisphere). Pigs were fed *ad libitum* and had free access to water via nipple drinkers in all the production phases. The diets used during the weaner phase, the grower phase and the finisher phase were the same between the two seasons (Appendix A). Average daily feed intake (ADFI), average daily gain (ADG), and feed conversion ratio (FCR) were calculated for 9–14 weeks and 14 weeks to slaughter, separately.

### 2.7. Tissue Composition of Primal Cuts Using Dual-Energy X-ray Absorptiometry (DXA)

Dual-energy X-ray absorptiometry (DXA) has been validated to be able to accurately estimate muscle, fat and bone composition in pig primal cuts and carcasses [9]. For carcass composition measurements, 24 pigs (12 born-light and 12 born-normal pigs, balanced sex) from each mating season were selected from those were slaughtered at a similar body weight and on one calendar day (to suit the availability of the abattoir). The selection from pigs killed on one calendar day may have suppressed the age difference between the born-light and born-normal pigs used for the DXA scan. After slaughter, the left side of the carcass of the focal pigs was separated into primal cuts- shoulder, loin, belly and leg. Primal cuts were separately placed onto the DXA scanning platform (Hologic Discovery W DXA scanner) for scanning.

### 2.8. Statistical Analysis

Data on sow body conditions, total litter weight, piglet weight (average of each litter) and litter size were analysed using UNIVARIATE procedure with the main effects of mating season (summer vs. autumn) and parity of sows (defined as Parity 2, 3, 4, 5 and 6+). Data on rectal temperature, respiration rate, and number of skin injuries of gestating sows were analysed using Repeated Measures with the day of measurement as a within-subject factor and mating season as a between-subject factor. Farrowing rates of sows were analysed by Pearson’s *Chi*-squared analysis. Data on placental weight and piglet birth weight from focal new-born piglets were analysed for the main effects of mating season and parity of sows using sow’s ID (nested within mating season) as a random factor. Growth performance and carcass traits of progeny pigs were analysed using UNIVARIATE procedure with the main effects of mating season (summer vs. autumn), piglet birth weight class (born-light vs. born-normal) and the interaction. A pen (a group of 4 pigs) was used as an experimental unit for measuring growth performance between 9 and 14 weeks of age. An individual pig was used as an experimental unit for measuring carcass traits and growth performance between 14 weeks of age and slaughter. All analyses were conducted in SPSS (IBM SPSS Statistics for Windows, v25, Armonk, NY, USA). Continuous variables are presented as mean ± standard error (SE), and binominal data are reported as a percentage of distribution. Means were considered to be significantly different when *p* ≤ 0.05, and a trend was considered to exist when *p* ≤ 0.10. 

## 3. Results

### 3.1. Change of Body Condition during Gestation

Sows that were mated in summer were lighter on the second day of gestation (250 kg vs. 270 kg *p =* 0.003) but gained more weight by the 110th day of gestation than those mated in autumn (36 kg vs. 18 kg, *p <* 0.001) so that sow body weight at the 110th day of gestation was similar between the two mating seasons (Table 1). Backfat thickness measured on the 2nd day of gestation tended to be lower in sows mated in summer than in autumn (24.1 vs. 27.3 mm, *p =* 0.080), and the backfat thickness of sows mated in summer was significantly lower than those mated in autumn (23.2 vs. 25.7 mm, *p =* 0.033) by the 110th day of gestation. The change of backfat thickness between the 2nd and 110th day of gestation did not differ (*p =* 0.75) between sows mated in summer and autumn. The average daily feed intake during gestation was controlled to the same amount between the sows mated in summer and autumn.

### 3.2. Farrowing Outcomes

Sows with a similar average parity were compared between summer and autumn mating seasons for farrowing outcomes (Table 2). The sows mated in summer had a lower farrowing rate than those mated in autumn (75% vs. 89%, *p <* 0.001). The number of total born, born alive, stillborn and mummified piglets were all similar between the two mating seasons. Litter birth weights were lighter in the sows mated in summer than those mated in autumn (17.3 kg vs. 18.4 kg, *p =* 0.024 when only born-alive piglets were included; 18.5 kg vs. 19.6 kg, *p =* 0.044 when stillborn piglets were included). Average piglet birth weight was lighter for the sows mated in summer than those mated in autumn (1.39 kg vs. 1.55 kg, *p <* 0.001 (born-alive piglets only); 1.37 kg vs. 1.54 kg, *p <* 0.001 (stillborn piglets were included)). The percentage of born-alive pigs weighing less than 1.1 kg was greater in the sows mated in summer than in autumn (24.2% vs. 15.8%, *p <* 0.001).

### 3.3. Placental Weight of Focal Sows

The focal sows from which placental weights were measured had a similar average parity (Table 3). The number of total born, born-alive, still born and mummified piglets were also similar between the focal litters from the two mating seasons. Overall, the individual piglet birth weight increased with placetal weight at a reduced rate in both mating seasons (Quadratic response, both *p* < 0.001; Appendix A).The average placental weight was lower in the sows mated in summer than for those mated in autumn (0.26 kg vs. 0.31 kg, *p <* 0.001). The ratio between piglet birth weight and its matched placenta weight tended to be greater in the sows mated in summer than those mated in autumn (5.72 vs. 5.11, *p =* 0.054).

### 3.4. Physiological Signs of Heat Stress and Number of Scars on Sows

Sows mated in summer showed a higher respiration rate (Figure 3a) and rectal temperature (Figure 3b) during the first eight weeks of gestation compared with those mated in autumn (both mating season and gestation weeks interaction, *p <* 0.001). The number of fresh scars was significantly greater in the sows mated in summer than in those mated in autumn in the first three weeks of gestation (mating season and gestation weeks interaction, *p <* 0.001, Figure 3c).

### 3.5. Plasma IGF-1 Concentration of Newborn Piglets

New-born piglets from sows mated in summer and autumn had similar plasma IGF-1 concentrations (Figure 4). The born-light pigs had a markedly lower plasma IGF-1 concentration than born-normal piglets (50.3 vs. 88.6 ng/mL, *p <* 0.001), regardless of the mating seasons of sows. The interaction between mating season and progeny birth weight class was not significant.

### 3.6. Growth Performance from 9 to 14 Weeks of Age and from 14 Weeks of Age to 101 kg Live Weight

Progeny pigs were selected into a grower shed at 9 weeks of age with a similar body weight between mating seasons (Table 4). Born-light pigs had lower body weights than born-normal pigs (19.6 kg vs. 20.4 kg, *p <* 0.001) at selection. Between 9 and 14 weeks of age, pigs that were born to the sows mated in summer had a greater ADFI (1.48 kg/day vs. 1.35 kg/day, *p <* 0.001), grew faster (0.81 vs. 0.74 kg/day, *p <* 0.001), and were heavier at 14 weeks of age (48.2 kg vs. 45.8 kg, *p <* 0.001) than those born to the sows mated in autumn. The FCR was not affected by mating season or birth weight class. The ADFI, ADG and FCR did not differ significantly between born-light and born-normal pigs during 9–14 weeks of age. The interaction between mating season and birth weight class was not significant in those measurements.

Between 14 weeks of age and growth to 101 kg live weight, the pigs that were born to the sows mated in summer had a greater ADFI (2.81 kg/day vs. 2.49 kg/day, *p <* 0.001), ADG (1.18 kg/day vs. 1.07 kg/d, *p <* 0.001), and took fewer days to reach the 101 kg slaughter weight (143.5 days vs. 151.9 days, *p <* 0.001) than the pigs born to the sows mated in autumn. The interaction between mating season and birth weight class was significant for ADFI (*p =* 0.008), such that the born-light progeny pigs had lower ADFI than born-normal piglets among the progeny born to the sows mated in autumn but not in summer. Born-light pigs had lower ADG than born-normal pigs (1.02 kg/day vs. 1.13 kg/day, *p <* 0.001), regardless of mating seasons. The interaction between birth weight class and mating season was significant (*p =* 0.039) for ADG, such that the difference in ADG between born-light and born-normal pigs was smaller in the progeny born to the sows mated in summer than in those born to sows mated in autumn. The FCR was higher in the born-light progeny than in the born-normal ones (2.42 vs. 2.33, *p =* 0.025), whereas it was not affected by mating seasons. Born-light pigs took longer to reach 101 kg weight than born-normal pigs (149.8 days vs. 145.5 days, *p <* 0.001). End-point slaughter weight was controlled to be similar between mating seasons and birth weight classes.

### 3.7. Carcass Traits of Progeny

Hot standard carcass weight was similar for the progeny born to the sows mated in summer and autumn, which is an expected outcome after slaughtering at a fixed live weight (Table 5). Pigs born to the sows mated in summer tended to have greater backfat thickness than those born to the sows mated in autumn (14.9 mm vs. 14.2 mm, *p =* 0.060). Born-light progeny had a greater carcass backfat thickness than born-normal progeny at slaughter (15.0 mm vs. 14.0 mm, *p =* 0.003). The interaction between mating season and birth weight class on carcass backfat thickness tended to be significant (*p =* 0.073), indicating it was the born-light progeny (not the born-normal progeny) that had a greater backfat thickness when born to the sows mated in summer vs. autumn. Loin depth tended to be greater in born-light than the born-normal progeny that were born to the sows mated in autumn, but this effect was not seen in the progeny born to the sows mated in summer, as evidenced by a trend between mating season and birth weight class (*p =* 0.066). Neither mating season nor progeny birth weight class affected the dressing percentage. The head weights of progeny pigs were not affected by the mating season of sows or the birth weight class.

### 3.8. Tissue Composition of Primal Cuts and Half Carcasses

Pigs born to the sows mated in summer had greater loin fat% (39.8% vs. 34.2%, *p =* 0.023) than those born to the sows mated in autumn (Table 6). Born-light pigs tended to have greater loin fat% (39.4% vs. 34.6%, *p =* 0.054) than born-normal progeny. The interaction between mating season and birth weight class on loin fat% tended to be significant (*p =* 0.087), such that it tended to be only the born-light (not the born-normal) progeny that had increased loin fat% when born to the sows mated in summer vs. autumn. Pigs born to sows mated in summer tended to have less loin lean% (41.0% vs. 45.0%, *p =* 0.067) than those born to the sows mated in autumn. Born-light pigs tended to have less loin lean% (41.0% vs. 45.0%, *p =* 0.066) than born-normal counterparts. The interaction between mating season and birth weight class on loin lean% tended to be significant (*p =* 0.10), such that it was only the born-light (not the born-normal) progeny that had reduced loin lean% when born to the summer vs. autumn mated sows. Similarly, the fat: lean ratio was greater in the pigs born to the sows mated in summer (1.08 vs. 0.79, *p =* 0.024) than those born to the sow mated in autumn. Born-light pigs had a greater fat: lean ratio (1.07 vs. 0.80, *p =* 0.040) than born-normal counterparts. The interaction between mating season and birth weight class for fat: lean ratio tended to be significant (*p =* 0.058), such that the born-light but not the born-normal progeny tended to have a greater fat: lean ratio when born to the sows mated in summer vs. autumn. Pigs born to the sows mated in summer had less bone% (19.2% vs. 20.8%, *p <* 0.001) than the progeny born to the sows mated in autumn. Born-light piglets tended to have less bone% (19.7% vs. 20.3%, *p =* 0.061). The interaction between mating season and birth weight class on bone% tended to be significant (*p =* 0.095), such that born-light progeny had less bone% than born-normal progeny only when they were born to the sows mated in summer.

Tissue composition or fat: lean ratio in shoulders, bellies or legs was not affected by the mating season, birth weight class or their interaction. Fat% or lean% of the half carcasses were not affected by mating season, birth weight class or their interaction. Progeny born to the sows mated in summer had less bone% (19.6% vs. 20.0%, *p =* 0.041) in the half carcass than those born to the sows mated in autumn. The effect of birth weight class or the interaction between mating season and birth weight class was not significant on bone% of the half carcass.

## 4. Discussion

We had two important findings from the experiment, which supported our hypotheses. First, we found that the sows mated in summer produced piglets with lower mean birth weights and an increased percentage of born-light piglets (≤1.1 kg) than those mated in autumn (24.2% vs. 15.8%). Second, the progeny born-light (but not the born-normal) from sows mated in summer had greater carcass backfat thickness (15.6 mm vs. 14.4 mm for the born-light progeny born to sows mated in summer vs. autumn) and loin fat % than the progeny born to the sows mated in autumn. Based on the above results, we estimate that the greater percentage of born-light progeny and the higher carcass backfat would increase the populational average of backfat of the finisher progeny from 14.06 mm to 14.39 mm (101 kg live weight at slaughter). This estimated magnitude of increase in the populational carcass backfat was similar as the multiple-year data collected from Western Australia (14.11 mm vs. 14.46 mm for progeny finished in summer vs. spring; corrected for carcass weight) [1]. Our research findings indicate that the greater proportion of born-light progeny from sows mated in summer contributes to the increased carcass fatness of pigs finished in spring.

The sows mated in summer had 16% lower average placental weight and 9% lower average piglet birth weight compared with those mated in autumn, suggesting an inferior placental and fetal development. The compromised placental and fetal development for the sows mated in summer is suspected to be partially associated with the hotter environmental conditions during early gestation. In the current experiment, the sows mated in summer experienced hotter conditions during the first half of gestation than those mated in autumn, as evidenced by greater rectal temperatures and respiration rates. Effects of gestational heat stress on pig fetal development remain unclear due to the limited number of published studies and small sample sizes often used. A recent study showed that exposing pregnant gilts to artificially hot conditions during the whole of gestation reduced average piglet birth weight by 16% without affecting the number of piglets born alive (published as an abstract) [10]. Another three climatically controlled experiments showed that birth weight was not affected by the hot conditions that occurred during the first- [2,11], second-half [2,11] or whole gestation [2,11,12]. More consistent evidence that gestational heat stress compromises placental and fetal development has been reported in sheep. For example, exposing pregnant sheep to hot conditions during mid and late gestation reduced placental weights by 54–58% and reduced fetal weights by 17–27% [13,14]. Another sheep study demonstrated that artificial gestational heat stress can compromise fetal development through a reduction in umbilical blood flow, net fetal glucose uptake, and net fetal oxygen uptake [15]. However, similar physiological evidence has yet to be obtained in pregnant sows. In the current study the ratio between piglet birth weight and placental weight was 12% greater in the sows mated in summer than autumn. The increased ratio may imply that the ability to support fetal growth per gram of placental tissue was upregulated, maybe as an effort to support fetal development when the placental growth was inhibited by summer conditions during early gestation. The physiological functions of placenta (eg: nutrients transport, blood flow, etc.) were not studied in the current experiment and warrant future investigations under climatic controlled conditions.

Unlike sheep, sows always produce numerous fetuses, and there has been no evidence from climatically controlled studies on whether the impacts of heat stress on gestating sows is homogenous on all the fetuses or are expressed as increased frequency of born-light progeny. Our experiment showed that in summer-mated sows, there was an increased proportion (from 15.8% to 24.2%) of born-light progeny (≤1.1 kg) in a litter. Blood IGF-1 (measured at 24 h post-farrowing) in either birth weight class did not differ between summer and autumn mating seasons, but it was markedly lower in the born-light than born-normal piglets. The reduced IGF-1 in blood and tissues has been reported previously in the neonatal piglets identified as being intrauterine growth restricted [16,17]. Combining the fact that a greater proportion of born-light pigs were produced from sows mated in summer, we postulate that there was a greater proportion of neonatal piglets with reduced IGF-1 born to the sows mated in summer. The strategies to improve the IGF-1 during the neonatal phase (eg: increasing colostrum intake [18]) should be investigated in the future given the important role of IGF-1 in muscle deposition in later life [19].

Another important finding is that only the born-light progeny from the sows mated in summer had greater backfat and loin fat composition when compared with the progeny born to the sows mated in autumn. The born-normal progeny born to the sows mated in summer had a similar carcass backfat thickness and carcass fat composition as the born-light and born-normal progeny from the sows mated in autumn. This finding suggests that it is the increased proportion of born-light progeny that contribute to the greater carcass fatness observed in spring in the Australian pig industry. The current seasonal comparison design did not allow us to distinguish if the greater carcass fatness of born-light progeny from sows mated in summer was due to the prenatal or postnatal environment. The progeny that were born to the sows mated in summer and finished in spring had an overall 9% and 12% greater feed intake for 9–14 weeks of age and 14 weeks of age to slaughter respectively compared with the progeny that born to sows mated in autumn and finished in summer. Similarly, a survey in an Australian piggery between 2006 and 2008 showed that ADFI of pigs finished in the spring season was 7% greater than that of pigs finished in summer [20]. Both prenatal and post-natal environmental conditions should be considered to explain the different overall feed intake between the finisher season in spring and summer. Inconsistent effects of gestational heat stress on feed intake of progeny have been reported in climatically controlled studies [11,21,22], with one of the studies reporting that gestational heat stress increased ADFI of progeny by 9% [11]. On the other hand, the inhibitive effect of postnatal heat stress on ADFI is more consistent. A meta-analysis showed that ADFI reduces by 32–78 g/d (depending on body weight of pigs) for each Celsius degree increase in environmental temperature above 20 °C [23]. We surmise that the increased carcass fatness of born-light progeny from the sows mated in summer may be exacerbated by the higher level of feed intake from 14 weeks of age to slaughter. Low birth weight pigs have reduced muscle deposition potential [24] and grow with greater adiposity [4,25], which may be associated with the reduced number of muscle fibers [26] and satellite cells [27]. In the scenario when a high feed intake is permitted (eg: a cooler environment during the grower/finisher phase), the born-light progeny may have to divert the extra amount of dietary energy intake towards fat deposition. By contrast, backfat thickness and loin fat composition of born-normal progeny (when slaughtered at a fixed live weight) were both similar between pigs born to the sows mated in summer and those born to sows mated in autumn despite their greater feed intake from 9 weeks of age to slaughter, suggesting that the born-normal piglets have a constant lean to fat gain ratio over a wider range of dietary energy intakes. Similarly, the disadvantage in muscle fiber numbers and lean tissue deposition rates were also reported in born-light sheep [28,29]. Strategies that can improve muscle deposition potential during finisher phase should be investigated in born-light progeny from sows mated in summer.

In addition to the hot conditions during early gestation, we also found other factors may potentially contribute to the reduced birth weights of progeny that were born to the sows mated in summer. The effects of these factors on fetal development warrant further investigations in controlled studies. First, the negative energy balance of sows during summer lactation may jeopardise the body conditions of sows at subsequent mating and have carry-over effects on fetal development. We found that sows mated in summer were 20 kg lighter at mating than those mated in autumn, which is likely to be a consequence of negative energy balance during summer lactation [30,31]. Similarly, in our previous lactation studies conducted in summer and autumn, we observed that sows lactating in summer had 15% lower ADFI and 33 kg less body weight at weaning than those lactating in autumn [32,33]. The restricted amount of energy allowance during gestation might have to be distributed towards maternal tissue recovery, thus nutrient allocation to the conceptus could be affected. Interestingly, we found that the sows mated in summer gained more body weight during gestation than those mated in autumn although the gestational nutrient intake was similar, which may be due to the greater body heat loss of the sows mated in autumn and then gestated over the winter season. Second, our experiment showed that the sows mated in summer had a high number of skin injuries in the first weeks post-mating, implying increased social stress during early gestation. Both rodent and human studies reported that maternal stress was associated with reduced fetal weights [34,35], so it is postulated that the increased social stress of sows post-mating in summer may be another factor causing a reduction in piglet birth weights.

The limitation of the current seasonal comparison experiment was that the data was only collected from one year. Ideally, the individual progeny birth weight, growth performance and body composition data collected from multiple years are required for validating the seasonal effects discovered in the current study.

## 5. Conclusions

Sows mated in summer produced an increased proportion of born-light piglets (≤1.1 kg), which may be associated with the hotter environmental conditions experienced during the first half of gestation. The born-light but not born-normal progeny from the sows mated in summer had an increased carcass backfat thickness and loin fat composition at slaughter, possibly due, at least in part, to the increased feed intake during a cooler grower/finisher phase, when compared with the progeny from sows mated in autumn. Future experiments are required to understand the exact mechanism for the increased proportion of born-light progeny born to the sows weaned and mated in hot conditions so that mitigation strategies can be developed.

## Figures and Tables

**Figure 1 animals-10-02080-f001:**
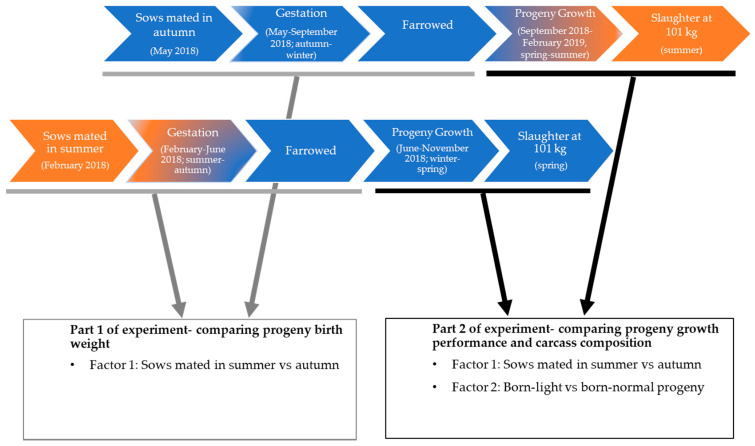
Flowchart of experimental design. In Part 1 of the experiment, physiological signs of thermal response during gestation and farrowing outcomes, including piglet birth weights, were compared between the sows mated in summer (*n =* 119 sows; February 2018, Australia) vs. autumn (*n =* 118 sows; May 2018, Australia). In Part 2 of the experiment, progeny of pigs that were born to the sows mated in summer and autumn were stratified as born-light (0.7–1.1 kg range) and born-normal (1.3–1.7 kg range) and grown to 101 kg live weight. Growth performance (from 9 weeks of age to 101 kg live weight) and carcass composition of progeny pigs were compared using two-way ANOVA for the effects of the mating season of sows (summer vs. autumn), progeny birth weight category (born-light vs. born-normal) and their interaction.

**Figure 2 animals-10-02080-f002:**
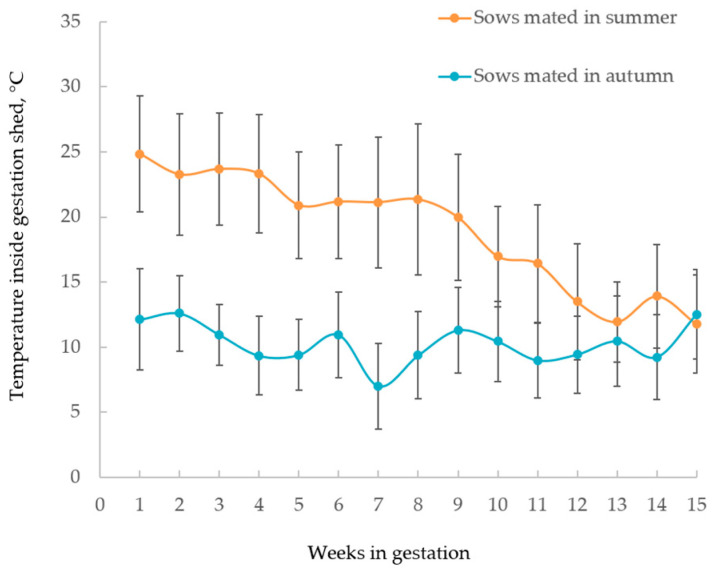
Temperature record of gestation shed (mean ± standard deviation).

**Figure 3 animals-10-02080-f003:**
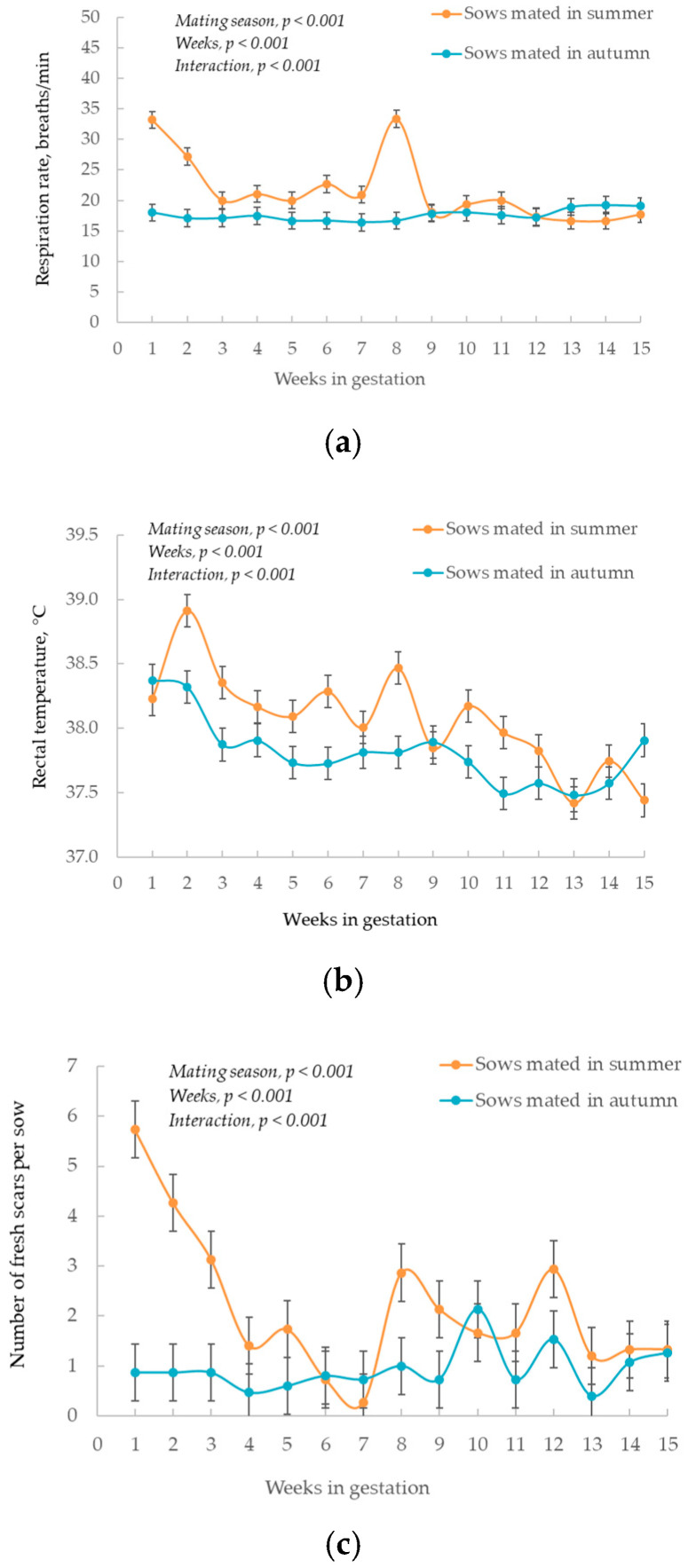
Respiration rate (**a**), rectal temperature (**b**), and skin injuries (**c**) of gestating sows mated in summer vs. autumn (*n =* 15 focal sows per mating season; values are expressed as mean ± standard error).

**Figure 4 animals-10-02080-f004:**
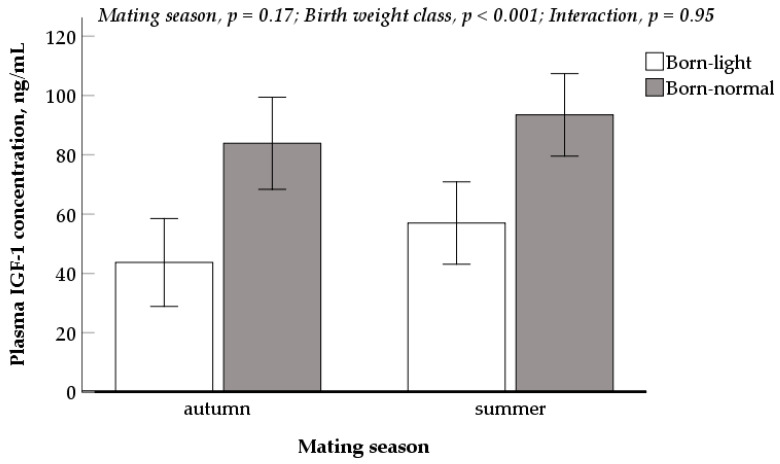
Plasma IGF-1 concentration (mean ± standard error) in newborn piglets from the sows mated in summer vs. autumn.

**Table 1 animals-10-02080-t001:** Gestational feed intake and body weight change of gestating sows mated in summer vs. autumn.

Variables	Mated in Summer (*n* = 15 Sows)	Mated in Autumn (*n* = 15 Sows)	SEM	*p*-Value
Parity of focal sows	4.1	4.1	0.39	1.00
Body weight (day 2), kg	250	270	4.1	0.003
Body weight (day 110), kg	286	288	3.7	0.74
Body weight gain, kg	36	18	2.2	<0.001
Backfat thickness (day 2), mm	24.1	27.3	1.23	0.080
Backfat thickness (day 110), mm	23.2	25.7	1.29	0.033
Backfat thickness change, kg	−1.1	−0.7	0.94	0.75
Average daily feed intake, kg	2.2	2.2	0.01	0.58

SEM, standard error of means for mating seasons.

**Table 2 animals-10-02080-t002:** Farrowing outcomes of sows mated in summer vs. autumn.

Variables	Mated in Summer (*n* = 119 Sows)	Mated in Autumn (*n* = 118 Sows)	SEM	*p*-Value
Parity of sows	3.7	3.8	0.13	0.83
Farrowing rate	75%	89%		<0.001
Number of piglets born	14.0	13.5	0.41	0.46
Number of born-alive piglets	12.6	12.5	0.29	0.73
Number of stillborn piglets	1.40	1.37	0.18	0.92
Number of mummified piglets	0.20	0.15	0.045	0.48
Litter birth weight (incl. stillborn), kg	18.5	19.6	0.040	0.044
Litter birth weight (born-alive), kg	17.3	18.4	0.36	0.024
Piglet birth weight (incl. stillborn), kg	1.37	1.54	0.026	<0.001
Piglet birth weight (born-alive), kg	1.39	1.55	0.026	<0.001
Proportion of piglets (born-alive) ≤1.1 kg, %	24.2	15.8	1.5	<0.001

SEM, standard error of means for mating seasons.

**Table 3 animals-10-02080-t003:** Placental weight of focal sows mated in summer vs. autumn.

Variables	Mated in Summer (*n* = 38 L)	Mated in Autumn (*n* = 38 L)	SEM	*p*-Value
Parity of sows	3.7	3.7	0.24	1.00
Number of piglets born	15.5	14.7	0.62	0.37
Number of born-alive piglets	13.8	12.9	0.51	0.25
Number of stillborn piglets	1.4	1.6	0.36	0.60
Number of mummified piglets	0.3	0.1	0.10	0.19
Placental weight, kg *	0.26	0.31	0.012	0.011
Piglet birth weight, kg *	1.37	1.48	0.038	0.040
Piglet: placental weight *	5.72	5.11	0.230	0.054

***** The total number of piglets born in a litter (15.1) was used as a co-variant. SEM, standard error of means for mating seasons.

**Table 4 animals-10-02080-t004:** Growth performance of focal progeny pigs born to sows mated in summer vs. autumn *****.

Variables	Progeny from Sows Mated in Summer		Progeny from Sows Mated in Autumn		SEM	*p*-Values		
	Born Light	Born Normal	Born Light	Born Normal		Mating Season	Birth Weight	Interaction
Body weight, 9 weeks, kg	19.6	20.4	19.6	20.4	0.27	0.96	0.005	0.92
ADFI, kg	1.45	1.50	1.35	1.36	0.029	<0.001	0.29	0.67
ADG, kg	0.79	0.82	0.74	0.73	0.015	<0.001	0.38	0.18
FCR	1.85	1.82	1.82	1.86	0.026	0.78	0.64	0.18
Body weight, 14 weeks, kg	47.2	49.2	45.5	46.2	0.56	<0.001	0.014	0.25
ADFI, kg	2.83	2.80	2.40	2.60	0.045	<0.001	0.070	0.008
ADG, kg	1.16	1.20	1.02	1.13	0.018	<0.001	<0.001	0.039
FCR	2.45	2.35	2.39	2.31	0.038	0.22	0.025	0.77
Days to slaughter	144.7	142.2	154.8	148.9	1.22	<0.001	<0.001	0.16
Body weight, slaughter, kg	101.0	101.4	102.1	102.7	0.69	0.079	0.47	0.90

***** Pigs were housed in a group of four between 9 weeks to 14 weeks of age (*n =* 14 pens per birth weight class × mating season treatment; equal number of pens per sex; *n =* 4 pigs per pen) then split and raised in individual pens from 14 weeks of age until reach slaughter weight (101 kg) (*n =* 56 pigs per birth weight class × mating season treatment). SEM, standard error of means for mating seasons × birth weight class.

**Table 5 animals-10-02080-t005:** Carcass traits (at 101 kg live weight) of focal progeny pigs born to sows mated in summer vs. autumn *.

Variables	Progeny from Sow Mated in Summer		Progeny from Sow Mated in Autumn		SEM	*p*-Values		
	Born Light (*n* = 56)	Born Normal (*n* = 56)	Born Light (*n* = 56)	Born Normal (*n* = 56)		Mating Season	Birth Weight	Interaction
Carcass weight, kg	77.0	77.4	77.5	78.1	0.58	0.29	0.41	0.83
Backfat, P2 site ^#^, mm	15.6	14.0	14.4	14.0	0.33	0.060	0.003	0.073
Loin depth ^#^, mm	53.6	54.1	54.4	52.2	0.70	0.47	0.26	0.068
Head weight ^†^, kg	6.9	6.9	7.0	7.0	0.06	0.27	0.89	0.41
Dressing, %	76.3	76.4	76.1	76.1	0.31	0.38	0.79	0.91

* Pigs were housed in a group of four between 9 weeks to 14 weeks of age (*n =* 14 pens per birth weight class × mating season treatment; equal number of pens per sex; *n =* 4 pigs per pen) then split and raised in individual pens from 14 weeks of age until reach slaughter weight (101 kg) (*n =* 56 pigs per birth weight class × mating season treatment).^#^ Carcass weight (77.5 kg) was used as a co-variate. ^†^ Live weight at slaughter (101.8 kg) was used as a co-variate. SEM, standard error of means for mating seasons × birth weight class.

**Table 6 animals-10-02080-t006:** Tissue composition (Dual-energy X-ray absorptiometry [DXA] method) of primal cuts and half carcasses of focal progeny pigs born to sows mated in summer vs. autumn.

Variables	Progeny from the Sow Mated in Summer		Progeny from the Sow Mated in Autumn		SEM	*p*-Values		
	Born Light (*n* = 12)	Born Normal (*n* = 12)	Born Light (*n* = 12)	Born Normal (*n* = 12)		Mating Season	Birth Weight	Interaction
Age at slaughter, days	137.5	136.4	140.2	139.3	0.44	<0.001	0.027	0.85
Live weight at slaughter, kg	100.9	100.0	99.3	100.4	0.84	0.49	0.89	0.25
HSCW, kg	76.4	76.1	76.0	77.2	0.66	0.65	0.49	0.28
Backfat thickness, mm	14.7	13.4	13.9	13.8	0.65	0.69	0.26	0.36
Shoulder								
Fat, %	32.2	29.8	30.2	28.9	1.49	0.35	0.22	0.70
Lean, %	47.1	49.2	49.0	50.0	1.42	0.34	0.28	0.71
Bone, %	20.7	21.0	20.8	21.0	0.21	0.85	0.20	0.78
Fat: Lean ratio	0.70	0.62	0.63	0.59	0.049	0.33	0.23	0.61
Loin								
Fat, %	44.2	35.4	34.5	34.0	2.30	0.023	0.054	0.087
Lean, %	37.2	44.8	44.8	45.2	2.08	0.067	0.066	0.10
Bone, %	20.7	21.0	20.8	21.0	0.21	0.85	0.20	0.78
Fat: Lean ratio	1.34	0.83	0.80	0.78	0.128	0.024	0.040	0.058
Belly								
Fat, %	38.1	35.6	36.3	37.2	1.89	0.94	0.39	0.66
Lean, %	48.2	50.1	49.0	49.6	1.89	0.94	0.50	0.75
Bone, %	13.7	14.3	13.9	14.0	0.34	0.99	0.28	0.51
Fat: Lean ratio	0.81	0.76	0.77	0.75	0.069	0.70	0.59	0.84
Leg								
Fat, %	24.2	21.2	23.4	24.4	1.66	0.49	0.54	0.23
Lean, %	54.9	57.8	55.8	54.7	1.58	0.50	0.57	0.22
Bone, %	20.9	21.0	20.8	20.9	0.20	0.73	0.56	0.86
Fat: Lean ratio	0.46	0.38	0.43	0.46	0.047	0.55	0.64	0.23
Half carcass								
Fat, %	32.9	28.7	29.2	29.2	1.55	0.29	0.19	0.19
Lean, %	47.8	51.4	50.9	50.9	1.45	0.39	0.22	0.22
Bone, %	19.3	19.8	20.0	20.0	0.19	0.041	0.15	0.14
Fat: Lean ratio	0.71	0.57	0.58	0.59	0.054	0.29	0.22	0.19

SEM, standard error of means for mating seasons × birth weight class.

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
