# Peer review of "The Greater Proportion of Born-Light Progeny from Sows Mated in Summer Contributes to Increased Carcass Fatness Observed in Spring"

_animals, 2020, doi:10.3390/ani10112080_

Round 1

Reviewer 1 Report

Animals 999332

Very well-written overall.  Good data set and interpretation for future efforts across the globe.

General comment: You postulate seasonal differences, but true seasonal effects can only be determined with replication of data within seasons over years.  Your study is a point in time (one year, one cycle of seasons) and I believe that, as you correctly indicate, additional research is needed. Thus, I believe acknowledging long-term evaluation of seasonal effects (replicated across years) would also be helpful as it will give more credence to direct relationships with observed seasonal variation in carcass composition (which have many years of data supporting seasonal variation).  This is a challenge faced by many groups.

L 225.  Your data do not show a significant backfat thickness change (less)  (Table 1) between d2 and d110 as written.  P=.75.

Table 2. SEM of proportion of piglets <1.1.  Check this value (0.015% does not seem possible when comparing two, population-based measures with max of 119 animals contributing to a given mean).

Table 3. Of the litters evaluated per period, and within the litters, you should have data to support whether placental weights collected were from light or normal weight category piglets.  Can you provide this data relative to weight class, as the data later show that only light weight pigs have the ‘detrimental’ effects within the seasonal effects.(ie. Interactions).  For the reader, placental efficiency may not be fully understood (e.g. which direction is more favorable) so this may warrant a note as well.  This shows up in lines 372-374 where you indicate greater efficiency indicates inferior uterine environment.    The simple ratios based on LSmeans are not close to the ratios presented in the table, must be a large covariate effect). 

L199  remove the word killed, replace with harvested or slaughtered.

L200.  Randomness of the pigs used in this portion of the study should be more clearly defined.  Appears to be multiple harvest days within a given season and birth weight class.  How was randomization performed across these differing harvest end points? 

L204 Statistical analyses.  Please clearly define your experimental units for gain between 9 and 14 weeks. Appears to be individual pig based on standard errors, rather than pen of pigs for ADFI and FCR.  If using pig as the exp unit in this period, please explain the implication.

L204 Statistical analyses.  Define the SEMs used within tables. Appear to be SEM of the Interaction means, but wish to make certain as the main effect SEMs would be different and attempting to identify main effect analyses within an interaction table are some of the outcomes that you are presenting.

Table 4. Interesting that days and body weight at slaughter are ‘round numbers’ vs earlier measurements (of weight in particular) that were 0.0 precision.  Is there a reason for differing precision of reporting?

Table 5. Head weight should likely be adjusted for live weight as a covariate.  This will likely clear up the numerical effects (Table 4) that seem to indicate a trend for mating season effects.  

Table 5.  Comment: Carcass data at P2 do not appear to differentiate loin influences on carcass lean nearly as well as the DXA. 

L368 – extra word (that) needs to be removed.

Line 371.  Your data suggest that in the AU system, increasing the proportion of light weight pigs by 8% within a season influences your country average carcass composition.  How much do you expect this small percentage of difference between seasons to influence the country average?   This was a central hypothesis that has not been addressed fully.   

Line 379 -  Do you suspect this was heat alone as you are suggesting or is it that they were also fighting (as indicated by more injuries) that lead to greater heat generation and increased respiration?  Likely both, but hard to pull apart.

Reviewer 2 Report

This manuscript has evaluated the pig production cycle contains seasonal variation in carcass fatness. This study found that sows were mated in summer they had an increased incidence of born-light progeny, which when finished in spring, had increased fatness. This finding provides a novel explanation for the seasonal variation of carcass fatness and sets a research direction for future mitigation strategies. This is an interesting topic. It could be published after few minor revisions.

  1. Line 24, there need a space after “≤”. Please go through the manuscript.
  2. Line 38, there need a space before and after “<”. Please go through the manuscript.
  3. Lines 121-122, please describe how dose the “average daily temperature” calculated with reference.
  4. Line 230, please add footnote for the table. Please check other tables, such as Table 6.
  5. Line 280, please add the different between the treatments.

Author Response

This manuscript has evaluated the pig production cycle contains seasonal variation in carcass fatness. This study found that sows were mated in summer they had an increased incidence of born-light progeny, which when finished in spring, had increased fatness. This finding provides a novel explanation for the seasonal variation of carcass fatness and sets a research direction for future mitigation strategies. This is an interesting topic. It could be published after few minor revisions.

Response: Thank you very much for the reviewing our manuscript. We have revised the manuscript and replied to your comments.

1. Line 24, there need a space after “≤”. Please go through the manuscript.

Response: Thank you. Spaces are added.

2. Line 38, there need a space before and after “<”. Please go through the manuscript.

Response: Thank you. Spaces are added.

3. Lines 121-122, please describe how dose the “average daily temperature” calculated with reference.

Response: We have added the sentence-“ Temperature loggers (DS1923 Hygrochron; OnSolution Pty, NSW, Australia) recorded the air temperature inside the sheds every hour. The average daily temperature was calculated based on the hourly recorded air temperature.”

4. Line 230, please add footnote for the table. Please check other tables, such as Table 6.

Response: Some footnotes are added.

5. Line 280, please add the different between the treatments.

Response: The difference between birth weight has been reported in the sentence.